# Regression Planning Networks

**Danfei Xu**
Stanford University

**Roberto Martín-Martín**
Stanford University

**De-An Huang**
Stanford University

**Yuke Zhu**
Stanford University
NVIDIA Research

**Silvio Savarese**
Stanford University

**Li Fei-Fei**
Stanford University

## Abstract

Recent learning-to-plan methods have shown promising results on planning directly from observation space. Yet, their ability to plan for long-horizon tasks is limited by the accuracy of the prediction model. On the other hand, classical symbolic planners show remarkable capabilities in solving long-horizon tasks, but they require predefined symbolic rules and symbolic states, restricting their real-world applicability. In this work, we combine the benefits of these two paradigms and propose a learning-to-plan method that can directly generate a long-term symbolic plan conditioned on high-dimensional observations. We borrow the idea of regression (backward) planning from classical planning literature and introduce Regression Planning Networks (RPN), a neural network architecture that plans backward starting at a task goal and generates a sequence of intermediate goals that reaches the current observation. We show that our model not only inherits many favorable traits from symbolic planning, e.g., the ability to solve previously unseen tasks, but also can learn from visual inputs in an end-to-end manner. We evaluate the capabilities of RPN in a grid world environment and a simulated 3D kitchen environment featuring complex visual scene and long task horizon, and show that it achieves near-optimal performance in completely new task instances.

## 1 Introduction

Performing real-world tasks such as cooking meals or assembling furniture requires an agent to determine long-term strategies. This is often formulated as a *planning* problem. In traditional AI literature, symbolic planners have shown remarkable capability in solving high-level reasoning problems by planning in human-interpretable symbolic spaces [1, 2]. However, classical symbolic planning methods typically abstract away perception with ground-truth symbols and rely on pre-defined planning domains to specify the causal effects of actions. These assumptions significantly restrict the applicability of these methods in real environments, where states are high-dimensional (e.g., color images) and it's tedious, if not impossible, to specify a detailed planning domain.

A solution to plan without relying on predefined action models and symbols is to learn to *plan from observations*. Recent works have shown that deep networks can capture the environment dynamics directly in the observation space [3–5] or a learned latent space [6–8]. With a learned dynamics model, these methods can plan a sequence of actions towards a desired goal through forward prediction. However, these learned models are far from accurate in long-term predictions due to the compounding errors over multiple steps. Moreover, due to the action-conditioned nature of these models, they are bound to use myopic sampling-based action selection for planning [4, 5]. Such strategy may be sufficient for simple short-horizon tasks, e.g., pushing an object to a location, but they fall short in tasks that involve high-level decision making over longer timescale, e.g., making a meal.

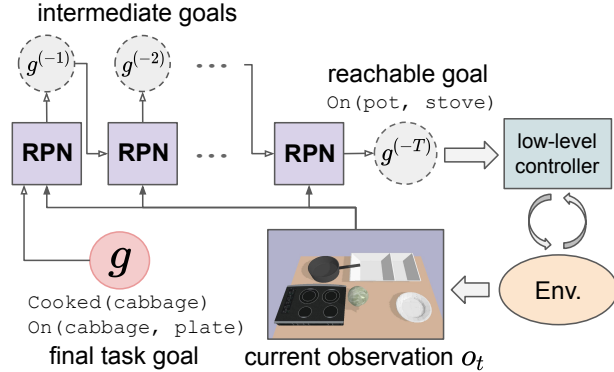

Figure 1: Regression (backward) planning with Regression Planning Networks (RPN): Starting from the final symbolic goal $g$, our learning-based planner iteratively predicts a sequence of intermediate goals conditioning on the current observation $o_t$ until it reaches a goal $g^{(-T)}$ that is reachable from the current state using a low-level controller.

In this work, we aim to combine the merits of planning from observation and the high-level reasoning ability and interpretability of classical planners. We propose a learning-to-plan method that can generate a long-term plan towards a symbolic task goal from high-dimensional observation inputs. As discussed above, the key challenge is that planning with either symbolic or observation space requires accurate forward models that are hard to obtain, namely symbolic planning domains and observation-space dynamics models. Instead, we propose to *plan backward* in a symbolic space *conditioning on the current observation*. Similar to forward planning, backward planning in symbolic space (formally known as regression planning [9, 10] or pre-image backchaining [11–13]) also relies on a planning domain to expand the search space starting from the final goal until the current state is reached. Our key insight is that by conditioning on the current observation, we can train a planner to directly predict a *single path* in the search space that connects the final goal to the current observation. The resulting plan is a sequence of intermediate goals that can be used to guide a low-level controller to interact with the environment and achieve the final task goal.

We present *Regression Planning Networks (RPN)*, a neural network architecture that learns to perform regression planning (backward planning) in a symbolic planning space conditioned on environment observations. Central to the architecture is a *precondition network* that takes as input the current observation and a symbolic goal and iteratively predicts a sequence of intermediate goals in reverse order. In addition, the architecture exploits the compositional structure of the symbolic space by modeling the dependencies among symbolic subgoals with a *dependency network*. Such dependency information can be used to decompose a complex task goal into simpler subgoals, an essential mechanism to learn complex plans and generalize to new task goals. Finally, we present an algorithm that combines these networks to perform regression planning and invoke low-level controllers for executing the plan in the environment. An overview of our method is illustrated in Fig. 1.

We train RPN with supervisions from task demonstration data. Each demonstration consists of a sequence of intermediate symbolic goals, their corresponding environment observations, and a final symbolic task goal. An advantage of our approach is that the trained RPN models can compose seen plans to solve novel tasks that are outside of the training dataset. As we show in the experiments, when trained to cook two dishes with less than three ingredients, RPN can plan for a three-course meal with more ingredients with near-optimal performance. In contrast, we observe that the performance of methods that lack the essential components of our RPN degrades significantly when facing new tasks. We demonstrate the capabilities of RPN in solving tasks in two domains: a grid world environment that illustrates the essential features of RPN, and a 3D kitchen environment where we tackle the challenges of longer-horizon tasks and increased complexity of visual observations.

## 2 Related Work

Although recent, there is a large body of prior works in learning to plan from observation. Methods in model-based RL [3–5, 7] have focused on building action-conditioned forward models and perform sampling-based planning. However, learning to make accurate predictions with high-dimensional observation is still challenging [3, 4, 6, 7], especially for long-horizon tasks. Recent works have proposed to learn structured latent representations for planning [8, 14, 15]. For example, Causal InfoGAN [8] learns a latent binary representation that can be used jointly with graph-planning algorithm. However, similar to model-based RL, learning such representations relies on reconstructing

the full input space, which can be difficult to scale to challenging visual domains. Instead, our method directly plans in a symbolic space, which allows more effective long-term planning and interpretability, while still taking high-dimensional observation as input.

Our work is also closely related to Universal Planning Networks [16], which propose to learn planning computation from expert demonstrations. However, their planning by gradient descent scheme is not ideal for non-differentiable symbolic action space, and they require detailed action trajectories as training labels, which is agent-specific and can be hard to obtain in the case of human demonstrations. Our method does not require an explicit action space and learns directly from high-level symbolic goal information, which makes it agent-agnostic and adaptive to different low-level controllers.

Our method is inspired by classical symbolic planning, especially a) goal-regression planning [9] (also known as pre-image backchaining [11–13]), where the planning process regresses from instead of progressing towards a goal, and b) the idea of partial-order planning [10, 17], where the ordering of the subgoals within a goal is exploited to reduce search complexity. However, these methods require (1) complete specification of a symbolic planning domain [1] and (2) the initial symbolic state either given or obtained from a highly accurate symbol grounding module [18, 19]; both can be hard to obtain for real-world task domains. In contrast, our method does not perform explicit symbolic grounding and can generate a plan directly from a high-dimensional observation and a task goal.

Our network architecture design is inspired by recursive networks for natural language syntax modeling [20, 21] and program induction [22, 23]. Given a goal and an environment observation, our RPN predicts a set of predecessor goals that need to be completed before achieving the goal. The regression planning process is then to apply RPN recursively by feeding the predicted goals back to the network until a predicted goal is reachable from the current observation.

## 3 Problem Definition and Preliminaries

### 3.1 Zero-shot Task Generalization with a Hierarchical Policy

The goal of zero-shot task generalization is to achieve task goals that are not seen during training [24–26]. Each task goal $g$ belongs to a set of valid goals $\mathcal{G}$. We consider an environment with transition probability $\mathcal{O} \times \mathcal{A} \times \mathcal{O} \rightarrow \mathbb{R}$, where $\mathcal{O}$ is a set of environment observations and $\mathcal{A}$ a set of primitive actions. Given a symbolic task goal $g$, the objective of an agent is to arrive at $o \in \mathcal{O}_g$, where $\mathcal{O}_g \subset \mathcal{O}$ is the set of observations where $g$ is satisfied. We adopt a hierarchical policy setup where given a final goal $g$ and the current observation $o_t$, a high-level policy $\mu : \mathcal{O} \times \mathcal{G} \rightarrow \mathcal{G}$ generates an intermediate goal $g' \in \mathcal{G}$, and a low-level policy $\pi : \mathcal{O} \times \mathcal{G} \rightarrow \mathcal{A}$ acts in the environment to achieve the intermediate goal. We assume a low-level policy can only perform short-horizon tasks. In this work, we focus on learning an effective high-level policy $\mu$ and assume the low-level policy can be either a pre-trained agent or a motion planner. For evaluation we consider a zero-shot generalization setup [25, 26] where only a subset of the task goals, $\mathcal{G}_{train} \subset \mathcal{G}$, is available during training, and the agent has to achieve a disjoint set of test task goals $\mathcal{G}_{test} \subset \mathcal{G}$, where $\mathcal{G}_{train} \cap \mathcal{G}_{test} = \emptyset$.

### 3.2 Regression Planning

In this work, we formulate the high-level policy $\mu$ as a learning-based regression planner. Goal-regression planning [9–13] is a class of symbolic planning algorithms, where the planning process runs backward from the goal instead of forward towards the goal. Given an initial symbolic state, a symbolic goal, and a planning domain that defines actions as their pre-conditions and post-effects (i.e., *action operators*), a regression planner starts by considering all action operators that might lead to the goal and in turn expand the search space by enumerating all preconditions of the action operators. The process repeats until the current symbolic state satisfies the preconditions of an operator. A plan is then a sequence of action operators that leads to the state from the goal.

An important distinction in our setup is that, because we do not assume access to these high-level action operators (from a symbolic planning domain) and the current symbolic state, we cannot perform explicitly such an exhaustive search process. Instead, our model learns to predict the preconditions that need to be satisfied in order to achieve a goal, conditioned on the current environment observation. Such ability enables our method to perform regression planning without explicit action operators, planning domain definition, or symbolic states.

We now define the essential concepts of regression planning adopted by our method. Following [12], we define goal $g \in \mathcal{G}$ as the conjunction of a set of logical *atoms*, each consists of a *predicate* and a list of object arguments, e.g., `On(pot, stove)`$\wedge\neg$`Clean(cabbage)`. We denote each atom in $g$ a **subgoal** $g_i$. A subgoal can also be viewed as a goal with a single atom. We define **preconditions** of a goal $g$ or a subgoal $g_i$ as another **intermediate goal** $g' \in \mathcal{G}$ that needs to be satisfied before attempting $g$ or $g_i$ conditioning on the environment observation. An intuitive example is that `In(cabbage, sink)` is a precondition of `Clean(cabbage)` if the cabbage is, e.g., on the table.

## 4   Method

Our primary contribution is to introduce a learning formulation of regression planning and propose Regression Planning Networks (RPN) as a solution. Here, we summarize the essential regression planning steps to be posed as learning problems, introduce the state representation used by our model, and explain our learning approach to solving regression planning.

**Subgoal Serialization:** The idea of subgoal serialization stems from partial order planning [10, 17], where a planning goal is broken into sub-goals, and the plans for each subgoal can be combined to reduce the search complexity. The challenge is to execute the subgoal plans in an order such that a plan does not undo an already achieved subgoal. The process of finding such orderings is called subgoal serialization [10]. Our method explicitly models the dependencies among subgoals and formulates subgoal serialization as a directed graph prediction problem (Sec. 4.1). This is an essential component for our method to learn to achieve complex goals and generalize to new goals.

**Precondition Prediction:**  Finding the predecessor goals (preconditions) that need to be satisfied before attempting to achieve another goal is an essential step in planning backward. As discussed in Sec. 3.2, symbolic regression planners rely on a set of high-level action operators defined in a planning domain to enumerate valid preconditions. The challenge here is to directly predict the preconditions of a goal given an environment observation without assuming a planning domain. We formulate the problem of predicting preconditions as a graph node classification problem in Sec. 4.2.

The overall regression planning process is then as follows: Given a task goal, we (1) decompose the final task goal into subgoals and find the optimal ordering of completing the subgoals (subgoal serialization), (2) predict the preconditions of each subgoal, and (3) set the preconditions as the final goal and repeat (1) and (2) recursively. We implement the process with a recursive neural network architecture and an algorithm that invokes the networks to perform regression planning (Sec. 4.4).

**Object-Centric Representation:** To bridge between the symbolic representation of the goals and the raw observations, we adopt an object-centric state representation [27–29]. The general form is that each object is represented as a continuous-valued feature vector extracted from the observation. We extend such representation to $n$-ary relationships among the objects, where each relationship has its corresponding feature, akin to a scene-graph feature representation [30]. We refer to objects and their $n$-ary relationship as *entities* and their features as *entity features*, $e$. Such factorization reflects that each goal atom $g_i$ indicates the *desired* symbolic state of an entity in the scene. For example, the goal `On(A, B)` indicates that the desired states of the binary entity `(A, B)` is A on top of B. We assume that either the environment observation is already in such entity-centric representations, or there exists a perception function $F$ that maps an observation $o$ to a set of entity features $e_t^i \in \mathbb{R}^D, i \in \{1...N\}$, where $N$ is the number of entities in an environment and $D$ is the feature dimension. As an example, $F$ can be a 2D object detector, and the features are simply the resulting image patches.

### 4.1   Learning Subgoal Serialization

We pose subgoal serialization as a learning problem. We say that a subgoal $g_i$ *depends on* subgoal $g_j$ if $g_j$ needs to be completed before attempting $g_i$. For example, `Clean(cabbage)` depends on `In(cabbage, sink)`. The process of *subgoal serialization* [10] is to find the optimal order to complete all subgoals by considering all dependencies among them. Following the taxonomy introduced by Korf *et al*. [10], we consider four cases: we say that a set of subgoals is *independent* if the subgoals can be completed in any order and *serializable* if they can be completed in a fixed order. Often a subset of the subgoals needs to be completed together, e.g., `Heated(pan)` and `On(stove)`, in which case these subgoals are called a *subgoal block* and $g$ is *block-serializable*. *Non-serializable* is a special case of *block-serializable* where the entire $g$ is a subgoal block.

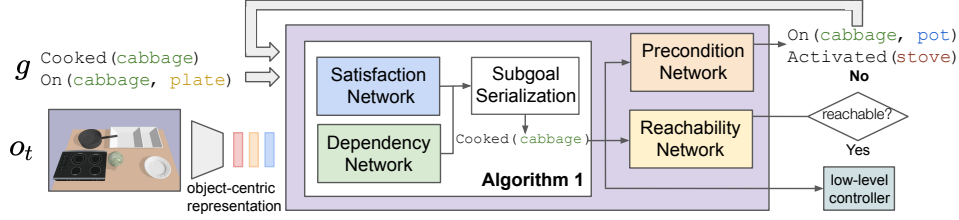

Figure 2: Given a task goal, $g$, and the current observation, $o_t$, RPN performs subgoal serialization (Algorithm 1) to find the highest priority subgoal and estimates whether the subgoal is reachable by one of the low-level controllers. If not, RPN predicts the preconditions of the subgoal and recursively uses them as new goal for the regression planning

To see how to pose subgoal serialization as a learning problem, we note that the dependencies among a set of subgoals can be viewed as a *directed graph*, where the nodes are individual subgoals and the directed edges are dependencies. Dependence and independence between a pair of subgoals can be expressed as a directed edge and its absence. We express subgoal block as a *complete subgraph* and likewise a non-serializable goal as a complete graph. The dependence between a subgoal block and a subgoal or another subgoal block can then be an edge between *any* node in the subgraph and an outside node or any node in another subgraph. For simplicity, we refer to both subgoals and subgoal blocks interchangeably from now on.

Now, we can formulate the subgoal serialization problem as a graph prediction problem. Concretely, given a goal $g = \{g_1, g_2, ..., g_K\}$ and the corresponding entity features $e_t^g = \{e_t^{g_1}, e_t^{g_2}, ...e_t^{g_K}\}$, our subgoal dependency network is then:

$$f_{dependency}(e_t^g, g) = \phi_\theta(\{[e_t^{g_i}, e_t^{g_j}, g_i, g_j]\}_{i,j=1}^K) = \{dep(g_i, g_j)\}_{i,j=1}^K, \tag{1}$$

where $dep(g_i, g_j) \in [0, 1]$ is a score indicating if $g_i$ depends on $g_j$. $\phi_\theta$ is a learnable network and $[\cdot, \cdot]$ is concatenation. We describe a subgoal serialization algorithm in Sec. 4.4.

## 4.2 Learning Precondition Prediction

We have discussed how to find the optimal order of completing a set of subgoals. The next step is to find the *precondition* of a subgoal or subgoal block, an essential step in planning backward. The preconditions of a subgoal is another goal that needs to be completed before attempting the subgoal at hand. To formulate it as a learning problem, we note that the subgoal and its preconditions may not share the same set of entities. For example, the precondition of `Clean(cabbage)` may be `In(cabbage, sink)`. Hence a subgoal may map to any subgoals grounded on any entities in the scene. To realize such intuition, we formulate the precondition problem as a *node classification problem* [31], where each node corresponds to a pair of goal predicate and entity in the scene. We consider three target classes, `True` and `False` corresponds to the logical state of the goal predicate, and a third `Null` class to indicate that the predicate is not part of the precondition. Concretely, given a goal or subgoal set $g = \{g_1, ..., g_K\}$ and all entity features $e_t$, the precondition network is then:

$$f_{precondition}(e_t, g) = \phi_\psi(\Delta(e_t, g)) = g^{(-1)}, \tag{2}$$

where $g^{(-1)}$ is the predicted precondition of $g$, and $\phi_\psi$ is a learnable network. Note that $g$ may only map to a subset of $e_t$. $\Delta$ fills the missing predicates with `Null` class before concatenating $e_t$ and $g$.

## 4.3 Learning Subgoal Satisfaction and Reachability

Subgoal serialization determines the order of completing subgoals, but some of the subgoals might have already been completed. Here we use a learnable module to determine if a subgoal is already satisfied. We formulate the subgoal satisfaction problem as a single entity classification problem because whether a subgoal is satisfied does not depend on any other entity features. Similarly, we use another module to determine if a subgoal is reachable by a low-level controller from the current observation. We note that the reachability of a subgoal by a low-level controller may depend on the state of other entities. For example, whether we can launch a grasping planner to fetch an apple from the fridge depends on if the fridge door is open. Hence we formulate it as a binary classification

problem conditioning on all entity features. Given a goal $g$ and its subgoals $\{g_1, ..., g_K\}$, the models can be expressed as:

$$f_{satisfied}(e_t^{g_i}, g_i) = \phi_\alpha([e_t^{g_i}, g_i]) = sat(g_i) \quad f_{reachable}(e_t, g) = \phi_\beta([e_t, g]) = rec(g), \quad (3)$$

where $sat(g_i) \in [0, 1]$ indicates if a subgoal $g_i \in g$ is satisfied, and $rec(g) \in [0, 1]$ indicates if the goal $g$ is reachable by a low-level controller given the current observation.

### 4.4 Regression Planning with RPN

Having described the essential components of RPN, we introduce an algorithm that invokes the network at inference time to generate a plan. Given the entity features $e_t$ and the final goal $g$, the first step is to serialize the subgoals. We start by finding all subgoals are unsatisfied with $f_{satisfied}(\cdot)$ and construct the input nodes for $f_{dependency}(\cdot)$, which in turn predicts a directed graph structure. Then we use the Bron-Kerbosch algorithm [32] to find all complete subgraphs and construct a DAG among all subgraphs. Finally, we use topological sorting to find the subgoal block that has the highest priority to be completed. The subgoal serialization subroutine is summarized in Algorithm 1. Given a subgoal, we first check if it is *reachable* by a low-level controller with

---

**Algorithm 1** SUBGOALSERIALIZATION

**Inputs:** Current entity features $e_t$, goal $g$
$v \leftarrow \emptyset$, $w \leftarrow \emptyset$
**for all** $g_i \in g$ **do**
    $sat(g_i) \leftarrow f_{satisfied}(e_t^{g_i}, g_i)$
    **if** $sat(g_i) < 0.5$ **then**
        $v \leftarrow v \cup \{g_i\}$, $w \leftarrow w \cup \{e_t^{g_i}\}$
    **end if**
**end for**
$depGraph \leftarrow \text{DiGraph}(f_{dependency}(w, v))$
$blockGraph \leftarrow \text{Bron-Kerbosch}(depGraph)$
$blockDAG \leftarrow \text{DAG}(blockGraph)$
$sortedBlocks \leftarrow \text{TopoSort}(blockDAG)$
**return** $g[sortedBlocks[-1]]$

---

$f_{reachable}(\cdot)$, and invoke the controller with the subgoal if it is deemed reachable. Otherwise $f_{precondition}(\cdot)$ is used to find the preconditions of the subgoal and set it as the new goal. The overall process is illustrated in Fig. 2 and is in addition summarized with an algorithm in Appendix.

### 4.5 Supervision and Training

**Supervision from demonstrations:** We parse the training labels from task demonstrations generated by a hard-coded expert. A task demonstration consists of a sequence of intermediate goals $\{g^{(0)}, ...g^{(T)}\}$ and the corresponding environment observations $\{o^{(0)}, ..., o^{(T)}\}$. In addition, we also assume the dependencies among the subgoals $\{g_0, ..g_N\}$ of a goal are given in the form of a directed graph. A detailed discussion on training supervision is included in Appendix.

**Training:** We train all sub-networks with full supervision. Due to the recursive nature of our architecture, a long planning sequence can be optimized in parallel by considering the intermediate goals and their preconditions independent of the planning history. More details is included in the Appendix.

## 5 Experiments

Our experiments aim to (1) illustrate the essential features of RPN, especially the effect of regression planning and subgoal serialization, (2) test whether RPN can achieve zero-shot generalization to new task instances, and (3) test whether RPN can directly learn from visual observation inputs. We evaluate our method on two environments: an illustrative Grid World environment [33] that dissects different aspects of the generalization challenges (Sec. 5.1), and a simulated Kitchen 3D domain (Sec. 5.2) that features complex visual scenes and long-horizon tasks in BulletPhysics [34].

We evaluate **RPN** including all components introduced in Sec. 4 to perform regression planning with the algorithm of Sec. 4.4 and compare the following baselines and ablation versions: 1) **E2E**, a reactive planner adopted from Pathak *et al.* [35] that learns to plan by imitating the expert trajectory. Because we do not assume a high-level action space, we train the planner to directly predict the next intermediate goal conditioning on the final goal and the current observation. The comparison to E2E is important to understand the effect of the inductive biases embedded in RPN. 2) **SS-only** shares the same network architecture as RPN, but instead of performing regression planning, it directly plans the next intermediate goal based on the highest-priority subgoal produced by Algorithm 1. This

baseline evaluates in isolation the capabilities of our proposed subgoal serialization to decompose complex task goal into simpler subgoals. Similarly, in 3) **RP-only** we replace subgoal serialization (Algorithm 1) with a single network, measuring the capabilities of the backward planning alone.

## 5.1  Grid World

In this environment we remove the complexity of learning the visual features and focus on comparing planning capabilities of RPN and the baselines. The environment is the 2D grid world built on [33] (see Table 1, left). The state space is factored into object-centric features, which consist of object types (door, key, etc), object state (e.g., door is open), object colors (six unique colors), and their locations relative to the agent. The goals are provided in a grounded symbolic expression as described in Sec. 3, e.g., Open(door_red)∧Open(door_blue). Both the expert demonstrator and the low-level controllers are $A^*$-based search algorithm. Further details on training and evaluation setup are in the Appendix. In the grid world we consider two domains:

**DoorKey:** Six pairs of doors and keys, where a locked door can only be unlocked by the key of the same color. Doors are randomly initialized to be locked or unlocked. The training tasks consist of opening $D = 2$ randomly selected doors (the other doors can be in any state). The evaluation tasks consist of opening $D \in \{4, 6\}$ doors, measuring the generalization capabilities of the methods to deal with new tasks composed of multiple instances of similar subtasks. The key to solving tasks involving more than two doors is to model opening each door as an independent subgoal.

**RoomGoal:** Six rooms connected to a central area by possibly locked and closed doors. The training data is evenly sampled from two tasks: **k-d** (key-door) is to open a randomly selected (possibly locked) door without getting into the room. **d-g** (door-goal) is to reach a tile by getting through a closed but unlocked door. In evaluation, the agent is asked to reach a specified tile by getting through a *locked* door (**k-d-g**), measuring the capabilities of the methods to compose plans learned from the two training tasks to form a longer unseen plan.

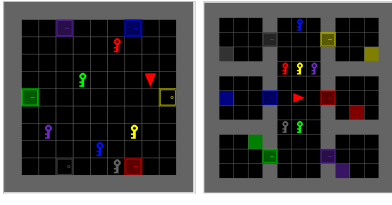

| Domain | DoorKey | | | RoomGoal | | |
|---|---|---|---|---|---|---|
| | Train | Eval | | Train | | Eval |
| Task | D=2 | D=4 | D=6 | k-d | d-g | k-d-g |
| E2E [35] | 81.2 | 1.2 | 0.0 | 100.0 | 100.0 | 3.2 |
| RP-only | 92.2 | 18.2 | 0.0 | 100.0 | 100.0 | **100.0** |
| SS-only | 99.7 | 46.0 | 21.1 | 99.9 | 100.0 | 7.8 |
| RPN | 99.1 | **91.9** | **64.3** | 98.7 | 99.9 | **98.8** |

Table 1: (Left) Sample initial states of DoorKey and RoomGoal domains; (Right) Results of DoorKey and RoomGoal reported in average success rate (percentage).

**Results:** The results of both domains are shown in Table 1, right. In *DoorKey*, all methods except E2E almost perfectly learn to reproduce the training tasks. The performance drops significantly for the three baselines when increasing the number of doors, $D$. RP-only degrades significantly for the inability to decompose the goal into independent parts, while the performance of SS-only degrades because, in addition to interpreting the goal, it also needs to determine if a key is needed and the color of the key to pick. However, it still achieves 21% success rate at $D = 6$. RPN maintains 64% success rate even for $D = 6$, although it has been trained with very few samples where all six doors are initialized as closed or locked. Most of the failures (21% of the episodes) are due to RPN not being able to predict any precondition while no subgoals are reachable (full error breakdown in Appendix).

In *RoomGoal* all methods almost perfectly learn the two training tasks. In particular, E2E achieves perfect performance, but it only achieves 3.2% success rate in the k-d-g long evaluation task. In contrast, both RP-only and RPN achieve optimal performance also on the k-d-g evaluation task, showing that our regression planning mechanism is enough to solve new tasks by composing learned plans, even when the planning steps connecting plans have never been observed during training.

## 5.2  Kitchen 3D

This environment features complex visual scenes and very long-horizon tasks composed of tabletop cooking and sorting subtasks. We test in this environment the full capabilities of each component in RPN, and whether the complete regression planning mechanism can solve previously unseen task instances without dropping performance while coping directly with high-dimensional visual inputs.

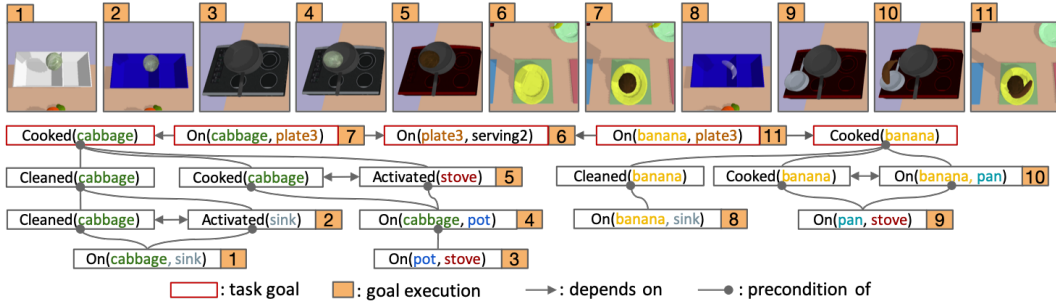

Figure 4: Visualization of RPN solving a sample cooking task with one dish and two ingredients ($I = 2$, $D = 1$): (Top) Visualization of the environment after a goal is achieved (zoom in to see details), and (Bottom) the regression planning trace generated by RPN. Additional video illustration is in the supplementary material.

**Cooking:** The task is for a robotic agent to prepare a meal consisting of a variable number of dishes, each involving a variety of ingredients and different cookwares. As shown in Fig. 3, the initial layout consists of a set of ingredients and plates randomly located at the center of the workspace surrounded by (1) a stove, (2) a sink, (3) two cookwares, and (4) three serving areas. There are two types of ingredients: fruits and vegetables, and six ingredients in total. An ingredient needs to be cleaned at the sink before cooking. An ingredient can be cooked by setting up the correct cookware at the stove, activating the stove, and placing the ingredient on the cookware. Fruits can only be cooked in the small pan and vegetables in the big pot.

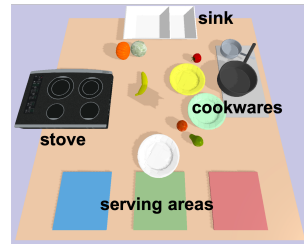

Figure 3: The Kitchen 3D environment. An agent (not shown) is tasked to prepare a meal with variable number of dishes and ingredients

The environment is simulated with [34]. A set of low-level controllers interact with objects to complete a subgoal, e.g., `On(tomato, sink)` invokes an RRT-based motion planner [36] to pick up the tomato and place it in the sink. For the object-centric representation, we assume access to object bounding boxes of the input image $o_t$ and use a CNN-based encoder to encode individual image patches to $e_t$. The encoder is trained end-to-end with the rest of the model. More details on network architectures and evaluation setup are available in the Appendix.

We focus on evaluating the ability to generalize to tasks that involve a different number of dishes ($D$) and ingredients ($I$). The training tasks are to cook randomly chosen $I = 3$ ingredients into $D = 2$ dishes. The evaluation tasks are to cook meals with $I \in \{2, ..., 6\}$ ingredients and $D \in \{1, .., 3\}$ dishes. In addition, cooking steps may vary depending on the order of cooking each ingredient, e.g., the agent has to set up the correct cookware before cooking an ingredient or turn on the stove/sink if these steps are not done from cooking the previous ingredient. In addition to the average task success rate, we also report the average sub-goal completion rate. For example, for an episode where 5 out of 6 ingredients is successfully prepared for a $I = 6$ task, the metric value would be $\frac{5}{6}$.

Table 2: Results of Kitchen 3D in average task success rate / average subgoal completion rate over 1000 evaluation episodes. All standard errors are less or equal to 0.5 and are thus omitted.

| | Train | Evaluation | | | | |
|---|---|---|---|---|---|---|
| Task | I=3, D=2 | I=2, D=1 | I=4, D=1 | I=4, D=3 | I=6, D=1 | I=6, D=3 |
| E2E [35] | 5.0 / 8.3 | 16.4 / 21.2 | 2.3 / 3.7 | 0.7 / 3.0 | 0.0 / <0.1 | 0.0 / <0.1 |
| RP-only | 70.3 / 83.4 | 67.1 / 77.4 | 47.0 / 71.7 | 27.9 / 64.1 | <0.1 / 23.9 | 0.0 / 22.9 |
| SS-only | 49.1 / 59.7 | 59.3 / 61.9 | 56.6 / 66.2 | 43.4 / 60.0 | 42.8 / 69.3 | 32.7 / 59.7 |
| RPN | **98.5 / 98.8** | **98.6 / 98.7** | **98.2 / 99.2** | **98.4 / 99.2** | **95.3 / 98.9** | **97.2 / 99.4** |

**Results:** As shown in Table 2, RPN is able to achieve near-optimal performance on all tasks, showing that our method achieves strong zero-shot generalization even with visual inputs. In comparison, E2E performs poorly on both training and evaluation tasks and RP-only achieves high accuracy on training tasks, but the performance degrades significantly as the generalization difficulty increases. This shows that the regression planning is effective in modeling long-term plans but generalize poorly

to new task goals. SS-only performs worse than RP-only in training tasks, but it is able to maintain reasonable performance across evaluation tasks by decomposing task goals to subgoals.

In Fig. 3, we visualize a sample planning trajectory generated by RPN on a two-ingredients one-dish task ($I = 2, D = 1$). RPN is able to resolve the optimal order of completing each subgoal. In this case, RPN chooses to cook cabbage first and then banana. Note that the steps for cooking a banana is different than that of cooking cabbage: the agent does not have to activate the stove and the sink, but it has to set a pan on the stove in addition to the pot because fruits can only be cooked with pan. RPN is able to correctly generate different plans for the two ingredients and complete the task.

Table 3: Results of RPN trained on $I = 3, D = 2$ tasks with different number of task instances $T$ and demonstrations per task $N$ and evaluated on $I = 6, D = 3$ tasks.

| Train Set | T=50 N=10 | T=100 N=10 | T=500 N=10 | T=1080 N=1 | T=1080 N=5 | T=1080 N=10 |
|---|---|---|---|---|---|---|
| RPN | 80.0 / 89.8 | 93.7 / 97.7 | 94.6 / 99.0 | 87.7 / 94.1 | 97.8 / 99.4 | 97.2 / 99.4 |

**Ablation study: generalization under limited data**. Here we evaluate RPN under limited training data. Specifically, we construct training datasets with reduced number of unique task instances $T$ (combinations of ingredients in a meal) and number of demonstrations per task instance $N$, and we evaluate the resulting RPN models on the most challenging $I = 6, D = 3$ tasks. As shown in Table 3, RPN generalizes well with both reduced $T$ and $N$. Notably, RPN is able to maintain a $\sim 95\%$ task success rate with 1/10 of all unique training task instances ($T = 100, N = 10$), showing that RPN can generalize to unseen goals and plans in a complex task domain.

# 6    Limitations and Future Works

**Partially-observable environments.** Because the current RPN framework assumes that the symbolic goals are grounded on the current observation, more architecture changes are required to accommodate goal specifications under POMDP. In addition, more principled approaches [19] may require extending RPN to explicitly reason about uncertainty of the state. We consider extending RPN to POMDP as an important future direction.

**Generalize to new objects and/or predicates.** We have shown preliminary results in Sec. 5 that RPN is able to generalize to new visual configurations of known predicates: An RPN model trained on tasks *I=3, D=2* has never seen a dish with more than two cooked ingredients, but it is able to generalize to tasks with four-ingredients dishes(*I=6, D=3*). Nonetheless, generalizing to arbitrary new objects and predicates remains a major challenge. For example, to plan for a goal Cooked(X) regardless of X requires understanding the invariant feature of Cooked. While possible in the current Kitchen3D environment, where predicates are rendered as simple change of texture hues, generalizing to more realistic scenarios would either require more diverse training data or explicitly learning disentangled representations [37] for the predicates.

**Learning new rules from a few demonstrations.** We have demonstrated that RPN can learn the implicit rules of a domain from thousands of video demonstrations. However, in a more realistic setting, we would like to have an agent that can learn new planning rules by only observing a few demonstrations, similar to [38]. A possible approach is to combine RPN with a meta-learning framework such as [39] to quickly adapt to a new domains.

**Generalized planning space.** In future works, we plan to extend the regression planning mechanism to more complex but structured planning spaces such as geometric planning [19] (e.g., including the object pose as part of a goal), enabling more fine-grained interface with the low-level controller. Another direction is to plan in a learned latent space instead of an explicit symbolic space. For example, we can learn compact representations of the entity features with existing representation learning methods [37] and train RPN to perform regression planning in the latent space.

# 7    Acknowledgement

This work has been partially supported by JD.com American Technologies Corporation ("JD") under the SAIL-JD AI Research Initiative. This article solely reflects the opinions and conclusions of its authors and not JD or any entity associated with JD.com.

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
