[Supplementary Material · RPN_Appendix.pdf]

# Regression Planning Networks: Apendix

## 1 Appendix

### 1.1 Architecture

Below we provide the details of our model size and architecture in the Kitchen 3D environment. The image encoder architecture is shared across all models. For Grid World, we use the same architecture but reduce all layer sizes by a factor of two. We use ReLU for activation. We train all models in all experiments with batch size of 128 and ADAM optimizer [1] with learning rate $= 1e - 3$ on a single GTX 1080 Ti GPU. We use a hold-out set for validation that runs every epoch. The models are implemented with PyTorch [2].

| | $f_{precondition}$ | $f_{dependency}$ | $f_{reachable}$ | $f_{satisfied}$ |
|---|---|---|---|---|
| RPN | MLP(128, 128, 128) | MLP(128, 128, 128) | MLP(128, 64, 64) | MLP(128, 128, 128) |
| RP-only | Same as RPN | N/A | Same as RPN | N/A |
| SS-only | N/A | Same as RPN | N/A | Same as RPN |
| E2E | MLP(256, 256, 256) | | | |
| Image encoder | [Conv2D(k=3, c=64), MaxPool(2, 2), Conv2D(k=3, c=128), MaxPool(2, 2), Conv2D(k=3, c=32), MaxPool(2, 2)] | | | |

### 1.2 Regression Planning Algorithm

Here we summarize the full regression planning algorithm (the subroutine SUBGOALSERIALIZATION is included in the main text). We set the maximum regression depth $M = 10$ for all experiments.

---
**Algorithm 1** REGRESSIONPLANNING
---
**Inputs:** Current entity features $e_t$, final goal $g$, maximum regression depth $M$
**Outputs:** Intermediate goal to be executed by a low-level controller.
$i \leftarrow 0$
**while** $i < M$ **do**
    $g' \leftarrow$ SUBGOALSERIALIZATION$(e_t, g)$
    $rec(g') \leftarrow f_{reachable}(e_t, g')$
    **if** $rec(g') > 0.5$ **then**
        **return** $g'$
    **end if**
    $g \leftarrow f_{precondition}(e_t, g')$
**end while**

---

### 1.3 Experiments Details

#### 1.3.1 Grid World

**Environment:** The environment is built on [3], where an agent moves around a 2D grid to interact with objects. There are three types of objects in our setup: door, key, and tile indicating a goal

location, and six unique colors for each object type. Doors can be in one of three states: (1) open, (2) close, (3) close and locked. A door can be only unlocked by the key of the same color, and a key can be used only once. Both the expert demonstrator and the low-level controllers are $A^*$-based search algorithm. A low-level controller can be invoked to execute subgoals, e.g., `Holding(key_red)`.

**Planning space:** We express task goals as conjunctive expressions such as `Open(door_red)` $\wedge$ `Open(door_blue)`. The symbolic planning space includes four unitary predicates: {`Open, Locked, Holding, On`}, where `Open` and `Locked` are for door-related goals. `Holding` is for picking up keys. `On` is for indicating a goal tile that the agent should reach in RoomGoal.

**State representation:** We factor the grid state information to object-centric features. Each feature is the concatenation of a set of one-hot vectors in the order of (1) object types (door, key, tile), (2) colors (6 in total), (3) object state (open, close, locked, holding), and (4) object location relative to the agent.

**Evaluation:** In the DoorKey domain, the task is to open $D$ out of 6 doors. We generate 5000 $D = 2$ training task demonstration trajectories by randomly sampling the state of the doors and the locations of keys and the agent. The evaluation tasks are opening $D \in \{4, 6\}$ doors. For RoomGoal, the training tasks are evenly sampled from (1) opening a possibly locked door without getting into the room (k-d) and (2) opening an unlocked door and reach a goal tile (d-g). We generate 2500 task demonstrations for each training tasks and evenly sample from these trajectories during training. All evaluation results are reported by running 1000 trials for each task instance.

### 1.3.2 Kitchen 3D

**Environment:** The 3D environment is built using Bullet Physics [4]. We use a disemboddied gripper for both gathering training data and evaluation. To minimize the effect of low-level controller and focus on evaluating the high-level planners, we assume the controllers are macros equipped with RRT motion planners, and the picking and placing movements are coded as setting and removing motion constraints between an object and the gripper. The placing poses are sampled randomly on the target placing surface with a collision checking algorithm provided by Bullet Physics. The controller in addition have an atomic action to activate the stove and the sink.

**Planning space:** The symbolic planning space includes four predicates: {`On, Cooked, Cleaned, Activated`}, where `On` indicates desired binary relationship between pairs of objects, and the others are unary predicates for specifying desired object states. A typical cooking goal specifies ingredients, mapping from ingredients to plates, and which serving area should a plate be placed.

**Rendering:** Because we directly take in image observation as input, the state changes of the objects (e.g., an ingredient is cooked, the stove is activated) should be reflected visually. All such visual state changes are implemented with swapping the mesh textures and / or setting the transparency of the texture. For example, cooking an ingredient darkens its texture, and cleaning an object makes the texture semi-transparent. We plan to extend such visual changes to be more realistic and include gradual changes of the state instead of instant changes in the future. We set the gripper to be invisible when rendering to minimize the effect of occlusion.

**State representation:** We render the scene into $320 \times 240$ RGB images with PyBullet's built-in renderer. For object-centric representation, we crop the images into image patches for individual object with object bounding boxes. We reshape the object bounding boxes to be the minimum enclosing square that can cover the full object and expand the box sizes by a factor of 1.1 to emulate an object detector. We resize all image crops to $24 \times 24$ and scale the pixels by $\frac{1}{255}$ before feeding to the image encoder (Sec. 1.1) and the rest of the network. The unary entity features are encodings of the individual image crops, and the binary entity features are the concatenation of encoding pairs.

**Scene setup:** For the scene setup, stove, sink, and the tray that initially holds cookwares have fixed locations. All ingredients and plates are initialized with random locations on the table. We have six ingredients in total, each fall into one of two types: fruits and vegetables. Fruits can only be cooked with the small pan, and vegetable can only be cooked with the big pot.

**Evaluation:** For training, we generate a total of 10800 trajectories for the task of cooking one dish with two randomly selected ingredients ($I = 2, D = 1$). Both the choice of plates and the serving areas are random. All evaluation results are reported with 1000 trials for each evaluation task. The standard error is reported on running 5 evaluation trials with different random seeds.

## 1.4   Obtaining Training Supervision

We use expert demonstration trajectories annotated with intermediate goal information as training data. We envision a few sources of such demonstrations. First, one can provide intermediate goal trajectories and use hard-coded policies to follow the intermediate goals as instructions to generate the corresponding environment observations. We use such setup in this work and generate training data on simple tasks, from which we train our RPN to generalize to more complex tasks. Second, we also intend to extend our work to use human demonstrations annotated with sub-task information (e.g., the data of the instructional video dataset [5]) as training data.

We include in Fig. 1 a sample intermediate goal trajectory and the subgoal-dependency information used to generated training data for the Kitchen3D environment ($I = 2, D = 1$). The precondition training label is parsed by tracing in the list of intermediate goals the subgoals that are part of the final goal recursively. Labels to learn subgoals dependency are provided as direct graphs as shown in the figure. *Satisfied* and *Reachable* labels can be directly parsed while stepping through the intermediate goal list in execution.

```
Goal: [on(plate/0, serving2), on(cabbage, plate/0), cooked(cabbage), on(banana, plate/0),
cooked(banana)]

Intermediate goals:
[on(cabbage, sink)]
[cleaned(cabbage), activated(sink)]
[on(pot, stove)]
[on(cabbage, pot)]
[cleaned(cabbage), cooked(cabbage), activated(stove)]
[on(plate/0, serving2)]
[on(cabbage, plate/0)]
[cleaned(banana), on(banana, sink)]
[on(pan, stove)]
[cleaned(banana), cooked(banana), on(banana, pan)]
[on(banana, plate/0)]

Dependencies:
cleaned(cabbage) <-> activated(sink)
cooked(cabbage) <-> activated(stove)
cooked(cabbage) -> cleaned(cabbage)
on(cabbage, plate/0) -> cooked(cabbage)
on(cabbage, plate/0) -> on(plate/0, serving2)
cleaned(banana) <-> on(banana, sink)
cooked(banana) <-> on(banana, pan)
cooked(banana) -> cleaned(banana)
on(banana, plate/0) -> cooked(banana)
on(banana, plate/0) -> on(plate/0, serving2)
```

Figure 1: A sample intermediate goal trajectory used to generated training data for the Kitchen 3D environment ($I = 2, D = 1$). Each row in the intermediate goals is a step to be completed by a low-level policy. Dependencies are global information and are used when applicable. Such type of annotation is commonly provided as labels in instructional video datasets, where video segments are annotated with step-by-step sub-task information.

## 1.5   Additional Results

### 1.5.1   DoorKey: Error Breakdown

Here we show a detailed error breakdown of the $D = 6$ evaluation tasks in the DoorKey environment.

Table 1: Error breakdown of $D = 6$ task in the Doorkey environment reported in percentage.

| Error Type | Success | Network | | | Environment | | |
|---|---|---|---|---|---|---|---|
| | | All Sat | No Prec | Max Iter | Controller | Bad Goal | Max Step |
| E2E | 0.0 | / | / | / | 0.3 | 92.6 | 7.1 |
| RP-only | 0.0 | 0.0 | 91.8 | 0.0 | 0.2 | 8.0 | 0.0 |
| SS-only | 21.1 | 0.0 | / | / | 0.0 | 78.9 | 0.0 |
| Ours | 64.3 | 0.9 | 21.3 | 8.7 | 0.1 | 1.4 | 3.3 |

We analyze two categories of errors: Environment and Network. Environment errors are errors occurred when the agent is interacting with the environment: **Controller** means that the low-level controller cannot find a valid path to reach a particular goal, e.g., all paths to reach a key is blocked. **Bad Goal** means that the goals predicted by the network are invalid, e.g., picking up a key that's already been used. **Max Step** is that the maximum of steps that the environment allows is reached.

Network errors are internal errors from components of our RPN network: **All Sat** means that the network incorrectly predicts that all subgoals are satisfied and exits prematurely. **No Prec** means that the network cannot predict any preconditions while none of the subgoals is reachable. **Max Iter** means that the regression planning process has reached the maximum number of steps ($M$ in Algorithm 1).

We see that the major source of error for E2E and SS-only is Bad Goal, i.e., the predicted goal is invalid to execute. We are able to catch these types of errors due to the simplicity of the grid world environment. However, this type of error may cause a low-level controller to behave unexpectedly in real-world tasks, causing security concerns. In contrast, RP-only and RPN both have very few such errors thanks to the robustness of our precondition networks. However, due to the inability to break a task goal into simpler parts, RP-only can easily make mistakes in the regression planning process, causing No Prec error. Finally, RPN is able to achieve 64% success rate while minimizing the errors occurred while interacting with the environment, highlighting a potential benefit of our system when deployed to real-world agents.