[Reviews · NeurIPS 2019]

Reviewer 1



Originality: The work proposes a novel method for regression planning via neural networks. Quality: The work is high quality and the authors have spent timing properly analyzing their method Clarity: The work is well written and techniques are well explained. Significance: This work as having potential to be impactful as the technique presented works well and aims at an important problem.

Reviewer 2



1. What is the role of learning in the tasks being studied in this paper? The paper shows results on planning problems. If I understand correctly, these problems can be solved exactly if appropriate state estimators are trained (to check on the state of different objects, such as, if the cabbage is cooked or is raw). Thus, to me it is not clear as to what is the role of learning in solving these problems? If the problem could be solved exactly using appropriate classical planners, why should learning be used here at all? The paper argues in its text that symbols need to be hand-defined in classical approaches, but as far as I understand, they have also been largely hand-crafted in the proposed learned approach. It will really help if: a) the authors can provide intuition as to what they hope to learn in the different networks they are learning? b) the authors can provide qualitative and quantitative evidence that the networks are learning something non-trivial? b) provide an explicit experimental comparison to a completely classical planning algorithm to solve the problem at hand? 2. The current experiments in the paper are geared towards showing that learning inside a planner scaffolding is better than without. However, the experimental setup chosen is in some sense biased towards problems that can be solved within a planner scaffolding. There are very clear symbols that need to be manipulated in order to get to the right solution, etc. Learning based solutions may be more attractive in settings where symbols may be fuzzily defined, hard to define, or hard to exactly extract out from raw sensory inputs. It will be great if the authors tried the proposed technique in such experimental settings, such as where data comes from more realistic settings.

Reviewer 3



Originality: The idea of integrating symbolic methods and neural networks to 1) learn better representations or 2) perform better search is not new. In this light, I would like to see a brief discussion of End-to-End Differentiable Proving, Rocktäschel and Riedel, 2017; this work also proposes a “neural” approach to perform backward chaining. (The authors *may* also consider going through the notation used in the paper to improve the clarity of the preliminaries section). Having said that, I think the work is still interesting, proposing a neat straightforward algorithm to perform planning. Quality: The paper is technically sound and is reasonably grounded in previous literature. Clarity: The paper is well-written and easy to understand (see below for minor comments on notation and typos) Significance: The work addresses an important challenge — that of building better reasoning systems — by integrating neural and symbolic methods to achieve best of both worlds. Apart from being a reasonable and grounded approach, RPNs also perform well experimentally on unseen goals and so, can be of general interest to the community.

[Author Response · NeurIPS 2019]

**(R1) Extension to POMDP domains.** We appreciate the insightful question. We are indeed planning to generalize
RPN to POMDP domains. The POMDP setting is challenging for both classical planning and modern learning-based
planning mainly because it requires *active sensing* steps [19]. Consider the example goal `InHand(Apple)` when the
apple is in a closed fridge. To learn to plan for such a goal within the RPN framework, we may assume that the
video demonstration includes sensing intermediate goals, e.g., a human demonstrator searches for the apple at a few
locations including the fridge. Given such demonstration data, we can train an RPN to predict these sensing goals (e.g.,
`IsOpen(Fridge)`) as the *preconditions* of achieving the final goal. However, more principled approaches [19] may
require extending RPN to explicitly reason about uncertainty of the state, which is out of the scope of this paper. We
will revise the current running example to avoid possible confusion as suggested by the reviewer.

**(R1) Task with goal `IsClose(Fridge) AND InHand(Apple)`.** We agree with the reviewer that this example presents
a challenging case, where solving the precondition `IsOpen(Fridge)` of the final goal `InHand(Apple)` undoes the
subgoal `IsClose(Fridge)`, which may cause the planner to oscillate between the two subgoals. Common solutions in
classical planning require considering the global planning solution to resolve the correct subgoal order [1]. However,
RPN does not assume a global planning setting. Solutions within the RPN framework can be either 1) annotate the
dependency IsClose(Fridge) -> InHand(Apple), which leads the planner to solve `InHand(Apple)` first regardless the
state of `IsClose(Fridge)`, or 2) consider the final task goal as a subgoal block instead of individual subgoals. We
will revise the paper to include a detailed discussion of this challenge.

**(R1) Source of supervision.** We note that the form and the granularity of the supervisions in this work are very similar
to that of instructional videos [36], which are extensively used in the video understanding community. Our eventual
goal is to scale RPNs to real-world tasks by learning from these instructional video datasets.

**(R2) Role of learning and difference from classical symbolic planners.** While RPN is inspired by classical planners,
the additional assumptions and the extra information required by classical planners prohibit a fair comparison to RPN.
As noted in the main text (line 5, 24, 91, 122-127), classical symbolic planners search for plans based on *planning*
*domains*, which define high-level actions as the symbolic rules of the task domains. The main challenge for symbolic
planners is thus not only to estimate entity states but even more to hand-define such rules for real-world domains at scale.
The key contribution of our paper is a novel neural network architecture that learns to plan from task demonstrations
without the need of a planning domain or explicit state estimation. Each component of the network directly takes
high-dimensional observations as input and encapsulates a part of the overall regression planning process (as described
in Sec. 4 of the paper). Furthermore, RPN works with an incomplete predicate set. For example, to plan in the kitchen
domain, a classical planner would require predicates such as `Graspable` and `Activatable` to define necessary rules,
whereas RPN learns to plan without these predicates (Appendix 3.2 includes a list of predicates used in the experiments).
We will revise the paper to further clarify these differences and the advantages of RPN.

**(R2) Application in more "realistic" settings.** First, we designed our evaluation environment to emulate a realistic
setting: our Kitchen 3D environment features visual scenes rendering meshes scanned from physical objects and full 6-
DoF object pose variations. Second, while the chosen task domains facilitate symbolic definitions, this type of domains
is widely adopted by prior works on real-world robot planning, especially in task and motion planning [12,13,19].
However, while these works assume access to symbolic planning domains, our paper presents a data-driven method that
learns to perform task planning without a planning domain. Furthermore, we note that because RPN represents the
planning steps with generic function approximators, we may use other representations that exhibit similar compositional
structures. As noted in the paper (line 345), we plan to extend RPN to work with hybrid symbolic-geometric task goals,
e.g., `On(A, B)` with a specified 6D pose. In such setting, the generated plans could be fed to an off-the-shelf motion
planner to enable task and motion planning. We will update the paper to highlight the relevance.

**(R4) Comparison to Rocktäschel and Riedel.** We thank the reviewer for the highly relevant pointer. Rocktäschel and
Riedel [2] proposes a neural theorem prover (NTP) that works with incomplete knowledge bases: it uses distributed
symbolic representations to resolve non-exact symbol matches and derive new logical rules. As noted by the reviewer,
NTP is related to our approach in that, similar to our regression planning mechanism, it employs a learnable backward-
chaining algorithm that recursively decomposes a proving goal into subgoals, and derives new proving states until the
terms in the state are supported by facts in the knowledge base. Besides the difference in the observation spaces, the
main conceptual difference between NTP and RPN is that the goal of NTP is to predict if a statement is true, whereas
RPN focuses on the correctness of the proof (plan) – whether the next intermediate goal in the plan is reachable and can
ultimately lead to the final goal. Hence, in principle, NTP may derive the correct answer from a wrong proof, whereas a
wrong plan (proof) would lead to failure in robotic tasks. On the other hand, RPN and NTP are complementary in many
aspects. For example, RPN may employ a distributed symbolic representation to handle task goals specified with an
open vocabulary. We will include a discussion on the relevance and will consider the notations in [2].

**(R4) Details.** $e_t$ denotes the entity features at time step $t$. We will incorporate the comments in the next revision.

[1] D. Chapman, "Planning for conjunctive goals," *Artificial intelligence*, vol. 32, no. 3, pp. 333–377, 1987.
[2] T. Rocktäschel and S. Riedel, "End-to-end differentiable proving," in *Advances in Neural Information Processing Systems*, pp. 3788–3800, 2017.


[Meta-Review · NeurIPS 2019]

This submission drew a great deal of discussion -- primarily on the point of the role of learning. All reviewers agreed that the approach had the potential to learn interesting, non-trivial things but did not feel the the current experiments demonstrated these effectively -- despite strong performance on the task. Some examples of questions that were not answered by the main draft but came up in the discussion: [Training Data] The training data provides edges in the dependency graph, subgoals, and predicate value -- image pairs. One question was whether the union of the seen dependency graph constituted the entire true underlying graph. Similarly, do all predicate-object pairs occur? These all get towards the question of "what is left to generalize to after training has finished?" [Generalization] Following up on that thought, where is evidence of learning generalized representations? Does the satisfcation network perform well on known predicates and objects in never-before-seen configurations? What about the reachability or dependency networks for combinations of known but never-before-paired subgoals? Does the precondition network understand that cooked(X) requires On(X,pot), Activated(stove) regardless of X? What if X is new? [Speed] One potential benefit of the precondition network is speed -- having learned useful / likely preconditions from the demonstrations. Given the demonstrations were generated using A* over the planning domain, has the precondition network learned to prune out ineffecient paths to goals? [Data Effiecency] While the demonstration counts were provided in the supplementary, it would be good to know how the performance of the approach varies with these. [Baselines] There are not really good baselines here to understand generalization of the learned components. The results certainly show that planning makes sense and that the decomposition of the networks is a good one, but not whether the learning is generalizing as described above. An ideal baseline would be something that can leverage the demonstrations and the planning algorithm without the possibility of generalizing. Something like a rule-learning algorithm might do the trick and would be limited to observed transitions. For inference, the satisfaction network could be used to estimate state. What is clear is that the proposed method can effectively learn from demonstrations without the need for explicitly learning the planning domain action rules like classical planners. This contribution is itself valuable even though the source for the detailed demonstrations may not yet be available. It seems likely that this paper would spur further work in this area. The pair of experimental settings make for convincing results. I do however recommend authors consider studying the questions posed above to demonstrate that the models have learned non-trivial things about the environments aside from those directly supervised by the demonstrations -- after all, it is somewhat clear that a large enough neural network with an infinite data generator like this could learn the set of binary and unary prediction tasks in a small synthetic environment.